# Exploring Neutrophil Heterogeneity and Plasticity in Health and Disease

**DOI:** 10.3390/biomedicines13030597

**Published:** 2025-03-01

**Authors:** Conny Gysemans, Mateson Beya, Erika Pedace, Chantal Mathieu

**Affiliations:** 1Leuven Diabetes Lab, Clinical and Experimental Endocrinology (CEE), Department of Chronic Diseases and Metabolism (CHROMETA), KU Leuven, 3000 Leuven, Belgium; beya.mateson@kuleuven.be (M.B.); chantal.mathieu@uzleuven.be (C.M.); 2Diabetes Unit, Department of Medicine, Surgery, and Neurosciences, University of Siena, 53100 Siena, Italy; erika.pedace@student.unisi.it; 3Fondazione Umberto Di Mario ONLUS c/o Toscana Life Science, 53100 Siena, Italy

**Keywords:** neutrophils, heterogeneity, plasticity, tissue microenvironment, therapeutics

## Abstract

Neutrophils, the most abundant polymorphonuclear leukocytes, are critical first responders to infection, and have historically been underappreciated in terms of their functional complexity within the immune response. Once viewed primarily as short-lived, innate immune cells with limited functional plasticity, recent research has illuminated their considerable heterogeneity and diverse functional roles, which extend beyond their involvement in steady-state immunity. This review seeks to provide an updated analysis of neutrophil development, maturation, heterogeneity, and plasticity, with a focus on how these characteristics influence immune modulation in both healthy and diseased tissues. Beginning with the origin of neutrophils, we explore their maturation into effector cells and their evolving roles in immune defense under homeostatic and disease-associated conditions. We then delve into their heterogeneity, discussing recent breakthroughs in neutrophil research that challenge the traditional view of neutrophils as a uniform population. We address the significant advances that have been made in identifying distinct neutrophil subsets, the emerging complexities of their plasticity, and the challenges that remain in fully understanding their functional diversity. Finally, we highlight future directions and opportunities for continued exploration in this rapidly advancing field, shedding light on how these insights could open new avenues for therapeutic interventions.

## 1. Introduction

Neutrophils are the most common innate leukocyte population in the human circulation, accounting for 50–70% of circulating leukocytes. Every day, roughly 100 billion neutrophils are generated from progenitor cells in the bone marrow, and serve as first responders to infection and inflammation. Traditionally recognized as key players in the innate immune system, neutrophils were once thought to be a homogeneous population. However, accumulating evidence suggests that these frontline defenders exhibit diverse functions, including pathogen elimination, tissue homeostasis maintenance, and modulation of inflammation, with their roles varying depending on the steady-state or pathological conditions they encounter [1,2]. This supports the idea that neutrophils form a heterogenous community, consequently igniting the interest of researchers in trying to classify neutrophil subsets [1]. Understanding the role and development of these distinct clusters of neutrophils could be advantageous for selective targeting of specific subpopulations that may contribute to disease pathogenesis [1]. The concept of a heterogenous neutrophil population has been studied in different contexts, such as inflammation/repair, autoimmunity, and cancer. However, there is still no unified understanding or agreement across these fields about neutrophil subsets [1]. In this review, we will explore the evidence for distinct neutrophil populations by examining their development from immature immune cells to fully differentiated cells, their various effector functions, and the growing body of research highlighting their remarkable plasticity.

## 2. Neutrophil Development and Lifespan

Neutrophils develop in the bone marrow, where they originate from CD34^+^ multipotent hematopoietic stem cells (HSCs) through a mechanism known as “granulopoiesis” (Figure 1). This process includes distinct maturation stages, during which neutrophils acquire key characteristics before becoming fully functional effector cells [1,2,3,4]. Overall, neutrophil ontogeny can be divided into mitotic and postmitotic phases, based on the stages of cell division and maturation [5]. The mitotic phase is where multipotent HSCs differentiate into common myeloid progenitor cells (CMPs), which in turn give rise to granulocyte-macrophage progenitor cells (GMPs) [5]. In humans, GMPs are lineage-committed progenitors characterized by Lin^−^c-Kit^+^CD34^hi^CD16/32^hi^, that give rise to typical GM colonies when cultured with methylcellulose containing steel factor, Flt-3 ligand (Flt-3L), interleukin (IL)-11, IL-3, GM-colony stimulating factor (GM-CSF), erythropoietin, and thrombopoietin [6]. In mice, equivalent stages involve Lin^−^Sca-1^+^c-Kit^+^ cells differentiating into CMPs and GMPs, marked by the expression of c-Kit, CD34, and CD16/32. During the mitotic phase, human pro-neutrophils (proNeus), which originate from GMPs, undergo a key decision in lineage commitment, differentiating into either myeloblasts, expressing c-Kit and CD34, or promyelocytes, expressing CD33 and CD13 [3,5]. In mice, myeloblasts and promyelocytes similarly express c-Kit and CD34, with CD34 expression being progressively downregulated as they further mature. The latter cells also express Ly6G, a marker that becomes more prominent in later stages. The transcription factor growth factor independence 1 (Gfi1) plays a central role in this preliminary conditioning, ensuring the cell is well equipped for its maturation and effector functions [5]. Gfi1-null mice are deficient in normal neutrophils, and instead accumulate an atypical population of Gr1^+^/CD11b^+^ cells, which display traits of both neutrophils and macrophages [7]. Gfi1 functions as a regulator of G-CSF/G-CSFR signaling in myeloid cells [8]. ProNeus exhibit plasticity to some extent and, in the absence of G-CSF, can evolve into monocytes rather than neutrophils [3,5].

During the lineage determination stage, which occurs at the transition between the mitotic and postmitotic phases, proNeus differentiate into pre-neutrophils (preNeus). Subsequently, preNeus, immature neutrophils (immNeus), and mature neutrophils (maNeus) undergo functional maturation, a process in which the progenitor cells gain the ability to perform their effector functions (see Section 4). Here, preNeus differentiate into myelocytes and, subsequently, metamyelocytes, exhibiting a kidney-shaped nucleus. In humans, myelocytes and metamyelocytes are defined by expression of CD15, CD11b, CD16, and the appearance of secondary granules (see Section 3.3). In mice, these cells express Ly6G and CD11b, and retain Ly6C, in contrast to their human counterparts. ImmNeus mature into band neutrophils, while maNeus are characterized by their hyper-segmented nuclei and fully functional granules [3,4,5]. In humans, band neutrophils (bandNeus) express CD16, CD11b, and CD10, while maNeus additionally express CD62L. In mice, the markers Ly6G and CD11b are used to identify band neutrophils, while the C-X-C chemokine receptor 2 (CXCR2) and CD62L are associated with maNeus.

After completion of the determination stage, neutrophils are released into the circulation, where they wander until recruited to sites of infection or inflammation [4]. The presence of pathogens at the infection site is crucial for extravasation [9]. Pathogens are detected either directly through pattern-recognition receptors, or by tissue-resident sentinel leukocytes, which release inflammatory mediators such as histamine, cysteinyl-leukotrienes, cytokines, and other factors that activate endothelial cells. During extravasation, activated endothelial cells upregulate the expression of various selectins, which bind to circulating neutrophils. This process occurs in three stages: tethering/rolling, activation, and firm adhesion, followed by trans-endothelial migration (reviewed by Nourshargh et al. (2014)) [10]. Although neutrophil recruitment mechanisms are fundamentally similar in mice and humans, both relying on selectins and integrins, the precise roles and timing of these molecules can differ between species. The widely accepted cascade paradigm of neutrophil recruitment may apply to many tissues, but recent findings suggest that recruitment mechanisms can vary significantly across different organs [11]. For instance, in organs such as the lung, liver, and kidney, the phenotype, morphology, and junctional composition of endothelial cells may differ, highlighting the need for revised, organ-specific models of inflammation (see Section 5.3). Additionally, neutrophils can migrate away from sites of inflammation and injury. However, the circumstances under which this “reverse migration” functions as a protective physiological response, as opposed to having pathophysiological implications, remain poorly understood [12].

Overall, neutrophils have a short lifespan in the circulation, ranging from 6 to 12 h [1]. Due to their short lifespan, these first responders are sensitive to the circadian rhythm, a 24 h cycle regulated by the suprachiasmatic nucleus region of the hypothalamus of the brain [13]. The suprachiasmatic nucleus is the master clock of the body, as it synchronizes the timing of circadian cycles in distinct tissues and organs, including immune cells, to maintain harmony throughout the whole body [14]. The circadian cycle helps to regulate the activity, function, and numbers of neutrophils, ensuring that the body maintains the appropriate amount of these first-line defenders at the appropriate times—such as during inflammation or when neutrophils need to be cleared. During the daytime, when the body is more susceptible to threats, neutrophils are highly present and exhibit higher activity. At night-time, neutrophils adapt to the body resting and recovering, consequently decreasing in number and activity [13]. Casanova-Acebes et al. (2013) report that clearance of neutrophils provides signals that modulate the physiology of the bone marrow. They identified a population of CD62L^lo^CXCR4^hi^ neutrophils that “aged” in the circulation and were eliminated at the end of the resting period in mice. Aged neutrophils infiltrated the bone marrow and promoted reductions in the size and function of the hematopoietic niche [15]. Overall, rhythmic fluctuations help the immune system to respond more effectively to infections [14]. Neutrophils have their own intrinsic circadian clocks which are controlled by the master clock in the suprachiasmatic nucleus. Their clocks are in tune with environmental changes and controlled by the core clock genes (e.g., CLOCK, BMAL1, PER, CRY), kinases, and phosphatases [14]. As organisms age, the circadian rhythms in various tissues tend to become less synchronized and lose their regularity, due to the expression of CLOCK, BMAL1, PER, and CRY becoming less robust. This impairment disrupts multiple physiological processes, including metabolism, sleep, and immune function, and results from the way in which aging impacts the function of the core circadian genes and proteins [14]. Since the circadian rhythm aids in regulating the timing of immune responses, any difference in the regular rhythm of immune cells, including neutrophils, can cause exaggerated immune activation and inflammation, which worsen aging-related issues [14].

## 3. Density-Based Heterogeneity in Neutrophils

Circulating neutrophils are traditionally isolated based on their density using gradient centrifugation techniques, a method that separates them into distinct subpopulations [16]. These subpopulations, referred to as high- and low-density neutrophils, exhibit significant differences in morphology, surface marker expression, and functionality. Hacbarth and Kajdacsy-Balla were the first to identify “low buoyant density neutrophils” in the peripheral blood mononuclear cell (PBMC) preparations of individuals with systemic lupus erythematosus, rheumatoid arthritis, and acute rheumatic fever [17]. The density-based heterogeneity of neutrophils is increasingly recognized as a crucial determinant of their functional plasticity in tissue homeostasis, regeneration, or chronic disease states [18].

### 3.1. High-Density Neutrophils

High-density neutrophils (HDNs) are the predominant population in the circulation of healthy individuals, characterized by a segmented nucleus and a granule-rich cytoplasm (see Section 3.3). In humans, HDNs express high levels of CD10, CD11b, CD16, and CD62L, while in mice, they express high levels Ly6G and CD11b. These maNeus are efficient in chemotaxis, and they play a central role in anti-microbial defense, acting as the first responders in the innate immune system to infection or tissue injury. Moreover, they are highly phagocytic and capable of producing reactive oxygen species (ROS) to clear pathogens and maintain tissue integrity. HDNs are implicated in disease development and progression. Their high density reflects their mature, quiescent state in the circulation [19]. The isolation and characterization of these cells has been described by Sagiv et al. (2015) [20].

### 3.2. Low-Density Neutrophils

Low-density neutrophils (LDNs) exhibit considerable heterogeneity in both their functional characteristics and morphological features [19]. They have been identified in different conditions, as well as in healthy individuals, accounting for approximately 5% of cells isolated from the PBMC density layer [17,21]. LDNs typically exhibit a banded nucleus, and a less granular cytoplasm compared to HDNs. This population can be further subdivided into immature (banded) and activated (hyper-segmented) subtypes, each with distinct characteristics. In humans, immature LDNs express low levels of CD16, high levels of CD33 and CD66b, and are often CD10^−^ or CD10lo [21]. Moreover, they lack major histocompatibility complex class II (MHC-II) and the costimulatory molecule CD86. In contrast, activated LDNs retain CD16 expression, but exhibit increased levels of activation markers such as CD64 and CD54. In mice, LDNs are identified by lower Ly6G expression and higher Ly6C levels, indicating a less mature or activated state [20,22]. Immature LDNs can be released from the bone marrow during “emergency granulopoiesis” in response to severe infection, inflammation, or stress. Alternatively, Sagiv et al. (2015) demonstrated that HDNs can transition into LDNs in a transforming growth factor (TGF)-β-dependent manner, losing anti-tumor activity and gaining immunosuppressive properties similar to those of immature granulocytic myeloid-derived suppressor cells (MDSCs) [20,22]. Activated LDNs, which are often associated with conditions like chronic inflammation, autoimmunity, and cancer, have been shown to exhibit enhanced neutrophil extracellular trap (NET) formation and pro-inflammatory cytokine production. These characteristics may contribute to tissue damage and disease progression [23]. Despite sharing similar granule compositions with HDNs, LDNs show a distinct functional profile, with varying levels of granule release and different responses to stimuli [21]. The potential cellular functions of LDNs remain a subject of considerable debate and controversy, and have been extensively reviewed [19].

### 3.3. Mechanisms Underlying Neutrophil Density Heterogeneity

Granule formation in neutrophils commences during the promyelocytic stage of granulopoiesis (Figure 1). Not only does the granule content play a key role in determining the density of the first responders, but it is also critical for their effector functions (see Section 4), enabling them to kill pathogens through a process referred to as degranulation [24]. When uncontrolled, degranulation can result in tissue injury and persistent inflammation. This underscores the importance of understanding how neutrophils manage their granule contents. Neutrophils synthesize and store three distinct types of granules: primary, secondary, and tertiary, which are categorized according to their developmental sequence, dimensions, morphology, and specific contents [25]. Primary or azurophilic granules, often referred to as peroxidase-positive granules due to their high content of myeloperoxidase (MPO), defensins, and other proteolytic enzymes, are synthesized during the promyelocytic stage from cis-Golgi. These granules are characterized by their spherical shape. As neutrophils progress to the myelocytic stage, secondary granules are produced. These granules include collagenase, gelatinase, lactoferrin, and lysozyme. Finally, tertiary granules, also known as gelatinase granules, form during the metamyelocytic stage. These granules are mainly composed of gelatinase, cathepsin, acyl transferase, and collagenase [24]. All granule subsets exhibit shared structural characteristics, including a phospholipid bilayer membrane and an intragranular matrix that harbors proteins earmarked for exocytosis or trafficking to the phagosome. Secretory vesicles, like granules, are specialized vesicles found in segmented neutrophils [26]. These vesicles are tightly regulated in their formation and release [27]. The presence of plasma proteins such as albumin within secretory vesicles indicates that they likely originate through the process of endocytosis, where material is internalized from the plasma membrane. Additionally, these endocytic vesicles serve as a reservoir of membrane-associated receptors essential for the initial stages of the neutrophil-driven inflammatory response [26,27].

HDNs are rich in primary, secondary, and tertiary granules, contributing to their higher density, while LDNs contain less granules. The granules in HDNs are crucial for their anti-microbial functions, whereas the reduced granule content in LDNs reflects their incomplete maturation or granule depletion. LDNs thus exhibit a diminished response to degranulation, except for the granule subset most readily triggered to undergo exocytosis, namely secretory vesicles. The density-based heterogeneity of neutrophils reflects their functional plasticity and adaptability to different physiological and pathological contexts. While HDNs are the primary effectors of anti-microbial defense, LDNs seem to play dual roles in immune regulation and pathology. Understanding the balance between these populations is critical for developing targeted therapies for diseases where neutrophil dysfunction is implicated [21].

## 4. Functional Heterogeneity in Neutrophils

Emerging evidence highlights the functional diversity of neutrophils in multiple disease contexts (see Section 6), suggesting they may contribute to both disease progression and resolution. In conditions such as infections and inflammatory and autoimmune diseases, as well as cancer, variations in neutrophil responses—including differences in phagocytosis, ROS production, degranulation, and NET formation—can influence the severity of symptoms, as well as the body’s ability to manage inflammation or infection (Figure 2). Neutrophils play a central role in regulating both innate and adaptive immunity through effector mechanisms such as phagocytosis, degranulation, and NET formation. Phagocytosis, the primary defense mechanism, involves the recognition and engulfment of pathogens or debris, resulting in the formation of a phagosome that matures into a phagolysosome. Within the phagolysosome, ROS and anti-microbial proteins eliminate the pathogen. This process is guided by various receptors, including opsonin receptors, which recognize antibodies and complement proteins, as well as non-opsonic receptors such as C-type lectins [28]. Degranulation is the process by which neutrophils release anti-microbial proteins from their granules and secretory vesicles into the phagosome or extracellular environment, to enhance pathogen destruction. However, certain bacteria have developed mechanisms to disrupt degranulation, enabling them to survive [29]. Additionally, neutrophils can eject NETs, structures built from decondensed chromatin and granule proteins. NETosis, the process of NET formation, can be triggered by inflammatory signals, immune complexes, bacteria, fungi, and protozoa (Figure 2) [30]. NETosis occurs in two forms: suicidal, which results in cell death and the release of chromatin; and vital, where NETs are released without cell death, thereby preserving their function [31]. NETosis can occur via nicotinamide adenine dinucleotide phosphate (NADPH)-dependent or NADPH-independent pathways. In the NADPH-dependent pathway, ROS generated by NADPH oxidase promote chromatin decondensation. In the NADPH-independent pathway, NET formation is triggered by calcium influx, which activates peptidyl arginine deiminase 4 (PAD4). PAD4 then citrullinates histones, leading to chromatin decondensation [29]. Beyond their role in pathogen defense, neutrophils modulate adaptive immunity, both directly and indirectly, through interactions with various immune cells, including T cells, B cells, dendritic cells, and monocytes [32,33,34,35]. For instance, neutrophils can behave as antigen-presenting cells, directly interacting with T cells [36]. This antigen presentation is enhanced by the expression of MHC-II and costimulatory molecules, particularly during inflammation driven by cytokines such as GM-CSF, interferon (IFN)-γ, IL-3, and tumor necrosis factor (TNF)-α. Matsushima et al. (2013) demonstrated that both immNeus and maNeus isolated from mouse bone marrow could differentiate into hybrid cells exhibiting dual characteristics of both neutrophils and dendritic cells when cultured with GM-CSF, but not with other tested growth factors. These hybrid cells expressed neutrophil markers (i.e., Ly6G, CD62L, CD24, CD11b, and CXCR2) alongside dendritic cell markers (i.e., CD11c, MHC-II, CD80, and CD86) [37]. Recently, Lad et al. (2024) used single-cell RNA sequencing to identify hybrid dendritic–neutrophil cells in both mouse and human glioblastomas, which were distinct from tumor-associated neutrophils (TANs) and originated from local precursors [38]. Neutrophils are also known to suppress T cell responses by secreting enzymes such as arginase-1 and ROS [39,40]. Furthermore, neutrophils influence T cell polarization (Figure 2), promoting either Th1 or Th2 responses, depending on the cytokines they produce. IL-12 secretion by neutrophils favors Th1 polarization, which enhances cell-mediated immunity, while IL-4 promotes Th2 polarization, promoting antibody-mediated immunity [41]. Additionally, neutrophil-derived granule proteins, such as azurocidin, participate in M1 macrophage polarization by monocytes, enhancing their pro-inflammatory response and phagocytic activity [35,42]. Thus, neutrophils bridge innate and adaptive immunity through their multifaceted roles in antigen presentation, cytokine production, and immune cell polarization [43]. However, despite these crucial functions, less is known about how neutrophil function varies in healthy individuals under steady-state conditions. Recently, Maskarinec et al. (2022) demonstrated that neutrophil functional variability is not random, but instead reflects a fixed phenotype unique to every individual, characterized by specific gene expression patterns [44]. Neutrophil functions such as phagocytosis, bacterial killing, ROS production, degranulation, and NET release all exhibit this functional heterogeneity [44]. Understanding this baseline variability in healthy subjects could provide valuable insights into how neutrophil functions are regulated and how they might be dysregulated in disease contexts.

## 5. Transcriptional and Phenotypic Heterogeneity in Neutrophils

Neutrophils are increasingly recognized as a heterogeneous population of immune cells, exhibiting distinct phenotypes and functions shaped by both cell-intrinsic (such as aging) and cell-extrinsic factors (such as stress). Their transcriptional and phenotypic variability begins in the bone marrow, and becomes more apparent once the cells enter circulation. Peripheral blood neutrophils display heterogeneity based on parameters such as maturity, buoyancy, cell-surface markers, function, and localization (Figure 3) [45,46]. These states are evident across both healthy conditions and various pathological settings, including autoimmune diseases, inflammatory disorders, and cancer. Recent advances, particularly single-cell RNA sequencing technologies, offer a promising approach for comprehensive and unbiased characterization of neutrophil states. Yet, most single-cell RNA analyses have been performed on PBMCs, with research lacking for most neutrophil populations. Moreover, low RNA content and high RNase activity in neutrophils may hinder accurate transcriptional profiling [47]. Their short lifespan and tendency for activation during isolation can complicate analysis even further. To address these challenges, researchers are developing optimized experimental protocols and modified data analysis pipelines. Beyond the circulation, accumulating evidence highlights the existence of tissue-based neutrophil populations [48,49]. These neutrophils, which can be either newly infiltrated or resident, acquire specialized phenotypes and functions dictated by the local tissue microenvironment (Figure 3). However, the lack of standardized nomenclature for neutrophils further complicates the understanding of their biology. This ambiguity raises the possibility that what are often considered biologically distinct populations of neutrophils, especially in disease contexts, may reflect the remarkable plasticity of neutrophils, rather than representing truly unique subsets. Understanding the complexities of neutrophil heterogeneity and plasticity is crucial for elucidating their multifaceted roles in health and disease, and for identifying novel therapeutic targets. Unified markers and criteria for subset identification are needed across studies.

### 5.1. Neutrophil Epigenetic and Transcriptional Regulation, from Bone Marrow to Circulation

Recent advances in transcriptomics, particularly single-cell RNA sequencing, have greatly enhanced our understanding of neutrophil heterogeneity, uncovering distinct maturation trajectories and functional diversity. Whole transcriptomic profiling and functional analyses have revealed that the formation of preNeus from GMPs relies on the transcription factor CCAAT/enhancer binding protein (C/EBPε). As these cells mature, they transition from a proliferative state to acquiring migratory and effector functions. Notably, preNeus expanded in response to microbial or tissue-induced stress, and immNeus were actively recruited to the periphery in tumor-bearing mice [50]. Kwok et al. (2018) identified an early committed progenitor within GMPs, termed proNeu1, which was responsible for the strict production of neutrophils, through combined single-cell transcriptomic and proteomic analyses. Furthermore, they demonstrated that the proNeu1 subset gave rise to an intermediate progenitor, proNeu2, which subsequently differentiated into downstream populations [51].

Re-analysis of the transcriptional regulatory changes in neutrophil subpopulations, defined by single-cell RNA sequencing across different samples, has revealed the involvement of a distinct cluster of transcription factors in neutrophil differentiation [52]. During neutrophil maturation, downregulated transcription factors include well-known differentiation factors (i.e, *Cebpa*, *Cebpz*, *Gata2*), as well as novel regulons (i.e., *Bclaf1*, *Hcfc1*). Transiently upregulated transcription factors in intermediate stages include *Cebpe* and *Ets1*, while terminal differentiation involves the upregulation of inflammatory regulons (i.e., *JunB*, *Nfe2l2*) and previously uncharacterized transcription factors (i.e., *Jund*, *Fos*). These findings align with stage-specific transcriptional networks that regulate neutrophil development [52]. A landmark study by Xie et al. (2020) described a neutrophil developmental branching pattern beyond the bone marrow. The authors profiled neutrophils from the bone marrow, spleen, and peripheral blood of healthy C57BL/6 mice, utilizing FACS sorting based on Gr1 antigen expression [53]. Their analysis identified eight distinct transcriptional clusters, five (G0–G4) in the bone marrow and three (G5a–G5c) in peripheral blood. Correlation analyses aligned the G0, G1, G2, G3, and G4 clusters with bone marrow GMPs, proNeus, preNeus, immNeus, and maNeus, respectively. While early maturation followed a single main branch, beginning with G2 cells in the bone marrow and progressing to G5 cells in peripheral tissues, G4 neutrophils bifurcated into two subsets, G5a and G5b subsets, which eventually converged into terminally differentiated G5c neutrophils. The G5c neutrophil population, characterized by the highest apoptotic scores and apoptotic proportions, marked the endpoint of neutrophil maturation. The G5a-c cells represented three transcriptionally distinct mature neutrophil subpopulations, each potentially pre-programmed with distinct functions. Gene ontology analysis revealed functional specialization along the maturation trajectory. Early neutrophils exhibited upregulation of genes associated with stem cell maintenance, early lineage commitment, ribonucleoprotein biogenesis, cytoplasmic translation, and cell cycle regulation, reflecting active proliferation and differentiation. In contrast, G3–G5 displayed enhanced expression of genes linked to phagocytosis, chemotaxis, and neutrophil activation, highlighting their key roles in immune response. This branching model suggests that, in addition to linear maturation, neutrophils can adopt specialized terminal states with distinct functional profiles. A recent study confirmed the previously characterized neutrophil subpopulations (G2–G4 and G5a–G5c) in both adult female and male C57BL/6 mice. However, the authors also revealed distinct transcriptional signatures of G5b neutrophils between sexes [54]. Grieshaber-Bouyer et al. (2021) also examined murine neutrophils from the bone marrow, spleen, and peripheral blood, identifying four clusters (P1–P4) based on Ly6G and CD11b antigen expression. The group introduced the concept of “neutrotime”, a continuum of neutrophil maturation from proNeus to immNeus and, eventually, maNeus. This linear trajectory suggested no branching into alternative phenotypes, but highlighted specific accumulation points throughout development. Bone marrow neutrophils primarily represented early stages, while the spleen housed more immNeus, and peripheral blood contained high numbers of maNeus. Gene expression profiles changed progressively along neutrotime, with metabolic and defense response genes predominating early, while toxin response, oxygen transport, and immune activation genes dominated the later developmental stages [55].

Applying single-cell RNA sequencing to human neutrophils revealed comparable transcriptional heterogeneity. Notably, Kwok et al. (2020) detected similar proNeu1, proNeu2, and preNeus in humans [51]. They also discovered that both proNeu1 and proNeu2 could give rise to maNeus. Additionally, a recent study identified a population of CD64^dim^CD66b^−^CD115^−^ cells in human bone marrow as neutrophil-committed progenitors [56]. In this study, the authors used RNA sequencing to profile neutrophil-committed progenitors, which include promyelocytes, myelocytes, metamyelocytes, banded cells, segmented neutrophils, and maNeus. Importantly, the different stages of neutrophils characterized by cell morphological features indeed exhibited distinct transcriptomic signatures, meaning that cell morphology and transcriptomics analyses could be integrated to precisely identify neutrophil stages. Moreover, Wigerblad et al. (2022) classified neutrophils purified from whole blood via immunomagnetic negative selection into four clusters (Nh0–Nh3). These clusters represented a continuum, ranging from relatively immNeus (Nh0), through an intermediate phenotype (Nh1), to two endpoints: one characterized by low transcriptional activity (Nh2), and the other by high expression of IFN-induced genes (Nh3). The Nh3 cluster, which expressed IFN marker genes, was transcriptionally distinct from the Nh2 state and has been consistently validated in both mouse models and humans. The transition from the less mature to more mature states in neutrophil development involved a complex regulatory network of transcription factors that play a crucial role in determining the fate of these cells [47]. Collectively, these studies emphasize the dynamic nature of neutrophil maturation and the transcriptional plasticity underpinning neutrophils’ functional diversity. By delineating distinct neutrophil subsets and their developmental pathways, transcriptomic approaches offer valuable insights into their roles in health and disease. Future research leveraging these techniques holds the potential to reveal novel therapeutic targets for intervention.

### 5.2. Neutrophil Reprogramming in Tissue Microenvironment

Neutrophils’ functions are dependent on their ability to be recruited and migrate to tissues and organs. Interestingly, neutrophils are recruited to tissues in a time-of-day-dependent manner, with the highest levels of recruitment occurring during specific phases of the circadian cycle (see Section 1). Furthermore, circulating neutrophils possess the capability to infiltrate a wide range of naïve tissues [57]. For instance, the spleen and lungs have been proposed as “reservoirs” for neutrophils, with their infiltration playing a critical role in maintaining tissue homeostasis. Neutrophils not only modulate the local environment, but also influence the function of resident cells [58]. Additionally, prostaglandin E2 and its downstream mediator, transglutaminase 2, in the lung have been shown to drive the neutrophil phenotype towards immune regulation and angiogenesis [59]. Upon entering naïve tissues, neutrophils seem to localize to distinct niches influenced by the tissue microenvironment, adapting their phenotype often independently of their initial maturation stage (Figure 3). Through protein, transcript, and epigenetic profiling, specific neutrophil subsets have been identified. These subsets, localized in healthy tissues, do not arise from pre-existing transcriptional programs in circulating neutrophils. Instead, they acquire tissue-specific properties that fulfill essential non-immune functions. For instance, a study found that lung-resident neutrophils expressed genes associated with vascular growth and repair (i.e., *Apelin*, *Adamts*, or *Vegfa*), while splenic neutrophils produced factors that enhanced immunoglobulin (Ig) production and maturation [60]. The pro-angiogenic population partially overlapped with a previously identified CD49d^+^ neutrophil subset described in the context of hypoxia in the lung [61]. In non-pathogenic lungs, neutrophils seem to accumulate at the periphery of veins, draining the lung parenchyma. Their tissue-specific reprogramming requires CXCL12/CXCR4 cues within niches [62]. These findings highlight that, despite their short residence in tissues, neutrophils can adapt to support tissue homeostasis and broader physiological functions. These specialized programs are unique to neutrophils and not shared by other leukocyte subsets in the same tissues, underscoring the specificity of microenvironmental programming. Despite limited transcriptional activity, neutrophils demonstrate great adaptability, maintaining tissue homeostasis through context-dependent reprogramming.

The functional plasticity of neutrophils becomes particularly evident in cancer, where the tumor microenvironment (TME) profoundly influences their behavior. Within the TME, TANs are categorized into two subtypes: N1 neutrophils, which exhibit anti-tumor activity, and N2 neutrophils, which promote tumor progression [63]. N1 neutrophils are characterized by higher expression of immune-activating cytokines and chemokines, contributing to their tumor-suppressive properties. Conversely, N2 neutrophils are driven by TGF-β signaling and associated with immunosuppression and angiogenesis, often exacerbating disease progression and leading to poor patient outcomes. N1 neutrophils, polarized by type I IFN, demonstrate the ability to eliminate tumor cells and inhibit tumor growth by recruiting and activating CD8^+^ T cells. Xue et al. (2022) described eleven neutrophil subsets in liver cancer, revealing that CCL4^+^ and PD-L1^+^ TANs, which recruit macrophages and suppress cytotoxic CD8^+^ T cells, respectively, were linked to unfavorable outcomes. These subsets could therefore serve as promising immunotherapy targets, either alone or in combination with immune checkpoint inhibition [64]. Despite the binary classification of N1 and N2 TANs, their roles remain complex and context-dependent, with subsets exhibiting both immunosuppressive properties, such as T cell suppression, and pro-angiogenic activities that facilitate tumor progression [65,66].

Recent evidence reveals that neutrophils infiltrating tumors can undergo extensive reprogramming, which varies by tumor type and reflects the heterogeneity of intra-tumoral niches. For example, in lung cancer, studies utilizing genetic mouse models and human patient data have identified at least six distinct transcriptional states of neutrophils. Some of these transcriptional profiles resembled those observed in healthy lung tissue, suggesting that the functional plasticity of neutrophils is highly adaptable and influenced by tissue-specific cues [67]. Both immNeus and maNeus entering tumors underwent permanent changes in gene expression, epigenetics, and proteins, leading them to adopt a distinct dcTRAIL-R1^+^ state. These reprogrammed neutrophils, which localized to the tumor’s glycolytic and hypoxic core, promoted angiogenesis and supported tumor growth. Similar changes were observed across various tumor types, suggesting that targeting this pathway could improve cancer immunotherapies [68]. Recently, a single-cell transcriptome of neutrophils from seventeen cancer types was performed. The neutrophils showed remarkable complexity, with ten distinct states, including those involved in inflammation, angiogenesis, and antigen presentation. The antigen-presenting state was linked to better survival in most cancers, and could be triggered by leucine metabolism and the modification of histone H3K27ac. These antigen-presenting neutrophils could activate both (neo-)antigen-specific and non-specific T cell responses. Additionally, delivering neutrophils or adjusting leucine intake improved immune responses and enhanced the effectiveness of immune checkpoint inhibitors in several mouse cancer models [69]. These findings collectively highlight the dual and context-dependent roles of TANs in cancer, emphasizing their potential as both targets for therapeutic intervention and contributors to the immune response against tumors.

### 5.3. Emergency Granulopoiesis by Extrinsic Cues

Emergency granulopoiesis is a finely tuned adaptive mechanism that enables the hematopoietic system in the bone marrow to swiftly meet the heightened demand for neutrophils during stress (e.g., infection, inflammation, or cancer) (Figure 3) [70]. This process involves an increased proliferation of early neutrophil progenitor cells in the bone marrow, coupled with an accelerated maturation once they exit the cell cycle. Importantly, while the fundamental stages of neutrophil differentiation within the bone marrow are preserved during emergency granulopoiesis, marked changes have been identified in the transitions between distinct neutrophil subpopulations [71]. Additionally, the spleen seems to play a role in emergency granulopoiesis, facilitating the swift replenishment of neutrophils lost during infections or other diseases [72]. Despite differences between the bone marrow niche and splenic stroma, neutrophil development in the spleen is thought to follow the same hierarchical process as in the bone marrow. In this way, the spleen can serve as a “reservoir” for myeloid cells, including monocytes and neutrophils, and can supply additional cells during immune challenges. Extrinsic cues that shape neutrophil heterogeneity during emergency granulopoiesis include signals from the tissue microenvironment, systemic factors such as hormones, immune mediators, and pathogen- or damage-associated signals, all of which can further regulate neutrophil activity and polarization [73]. However, it remains unclear whether these signals driving emergency granulopoiesis differ depending on whether they originate from infectious or non-infectious sources.

Hormones, immune mediators, or disease states (e.g., cancer) can create systemic changes influencing neutrophil heterogeneity. Inflammatory cytokines, such as IL-6 and TNF-α, can stimulate ongoing neutrophil production in rheumatoid arthritis and inflammatory bowel disease [74]. Diseases such as systemic lupus erythematosus often exhibit impaired granulopoiesis, partly driven by type I IFN [75]. Kwok et al. (2023) described disrupted granulopoiesis in sepsis, especially in patients with unfavorable outcomes. This patient population displayed an increase in IL1R2^+^ immNeus, elevated levels of emergency granulopoiesis markers such as C/EBP, and signal transducer and activator of transcription (STAT)-3-driven regulation across various infections [76]. In the cancer field, dysregulated granulopoiesis may also be driven by aberrant cytokine signaling (e.g., in chronic myeloid leukemia). During cancer progression, the release of cytokines such as G-CSF, GM-CSF, IL-3, IL-1β, and IL-6 into the TME can accelerate neutrophil production and promote their release from the bone marrow [76].

Signals including pathogen-associated molecular patterns (PAMPs) and damage-associated molecular patterns (DAMPs), recognized by PRRs such as toll-like receptors, can modulate neutrophil activation and polarization. Chronic inflammation in the context of rheumatoid arthritis can release DAMPs from damaged tissues, stimulating granulocyte production [77]. Moreover, PAMPs and DAMPs from gut microbiota and epithelial injury can trigger emergency granulopoiesis in inflammatory bowel disease [78]. Additionally, the TME is characterized by factors such as hypoxia, nutrient deprivation, cellular proliferation, and necrosis, all of which can trigger the release of DAMPs [79]. These DAMPs can then recruit and activate neutrophils, further fueling tumor-associated inflammation. In contrast, MDSCs generated through emergency granulopoiesis may play a role in immune evasion. MDSCs are immature myeloid cells that can be categorized into two main subtypes: granulocytic/polymorphonuclear MDSCs (PMN-MDSCs) and monocytic MDSCs (M-MDSCs). As their name implies, MDSCs play a crucial role in immune regulation by suppressing T cell activation and function [80]. While both cell types share common markers, PMN-MDSCs can be distinguished by their lower density and expression of the lectin-type oxidized LDL receptor-1 [81]. PMN-MDSCs display unique transcriptional profiles compared to normal density neutrophils (NDNs), with upregulation of genes associated with cell cycle progression, antigen presentation, and cytokine production, including TNF-α and IL-1β. Furthermore, these cells display elevated levels of endoplasmic reticulum stress markers, such as CAAT/enhancer binding protein homologous transcription factor (CHOP) and spliced forms of XBP-1 (sXBP1). Notably, increased expression of stress-induced β2-adrenergic receptors and ROS-mediated TNF-α-induced protein 8-like 2 (TIPE2) contributes to their immunosuppressive function, facilitating tumor immune escape and resistance to anti-tumor therapies [82]. Furthermore, PMN-MDSCs play a crucial role in the metastatic process. They can accumulate in premetastatic niches, where they exert immunosuppressive effects, induce extracellular matrix remodeling, and stimulate angiogenesis, thereby facilitating tumor cell dissemination [83].

## 6. Neutrophils as Therapeutic Targets in Disease Contexts

Neutrophils are now understood to exhibit diverse functional subsets that contribute to both protective and pathogenic responses in inflammation, autoimmunity, and cancer. Targeting not only their effector functions, but also more specific neutrophil subsets, may hold promise as a therapeutic strategy for modulating these diseases, offering a potential avenue for more precise and effective treatments.

### 6.1. Sepsis

Sepsis is a life-threatening condition characterized by a dysregulated host response to infection, frequently resulting in multi-organ dysfunction, and is the primary cause of death in intensive care units [84]. During sepsis, neutrophils exhibit an extended lifespan and are predominantly localized within the vasculature, where they release excessive levels of ROS, NETs, and pro-inflammatory cytokines. This overproduction significantly contributes to endothelial dysfunction and the progression of organ failure [85]. While inhibition of these effector pathways by selective inhibitors may have advantageous outcomes in sepsis (Figure 4), they are not selective to specific neutrophil subsets. Sun et al. (2022) described an increase in LDNs in the blood of septic subjects compared to healthy controls [86,87]. In this regard, Takizawa et al. (2021) identified extracellular cold-inducible RNA-binding protein (eCIRP) as a crucial mediator of LDNs. They showed that Ly6G^+^CD11b^hi^ LDNs were generated after recombinant murine CIRP stimulation in mice [88]. Therefore, targeting eCIRP using the C23 peptide or the miR-130b-3p mimic could offer potential therapeutic benefits for septic patients [89]. A recent study using single-cell and gene expression data of sepsis patients revealed that neutrophils were the second most prevalent cell type, and could be categorized into five distinct subsets: colony stimulating factor 3 receptor (CSF3R) neutrophils, matrix metalloproteinase-9 (MMP9) neutrophils, LYZ neutrophils, S100A9 neutrophils, and cystatin 3 (CST3) neutrophils. The proportion of CSF3R neutrophils and LYZ neutrophils was decreased in the sepsis group compared to healthy control group. As CSF3R activation is critical for neutrophil production, this could be an interesting target for treating individuals with sepsis.

### 6.2. Alzheimer’s Disease

Alzheimer’s disease (AD) is a neurodegenerative disorder marked by rapid progression that leads to significant memory decline. In individuals with AD, ROS production by blood neutrophils has been shown to be elevated, and neutrophils producing NETs have been detected in amyloid β aggregates within the brain [90]. These neutrophil effector functions can be targeted by ROS or PAD4 inhibitors (Figure 4). Furthermore, in the circulation of AD mice, six different subtypes of neutrophils have been identified according to distinct signature genes. Notably, a C-C motif chemokine receptor-like 2 (CCRL2)^+^ neutrophil subset, characterized as immNeus, was found to be increased in the blood and brains of AD mice compared to WT mice. CCRL2 is fundamental to neutrophil chemotaxis, and its increased expression may facilitate neutrophils’ migration into the brain, disturbing blood–brain barrier (BBB) integrity [91]. Moreover, mice lacking CCRL2 exhibited impaired neutrophil accumulation in inflamed joints, leading to protection against inflammatory arthritis [92]. Thus, CCRL2 emerges as promising therapeutic target to impede BBB damage.

### 6.3. Vasculitis

Vasculitis is an inflammatory disease affecting blood vessels, characterized by increased neutrophil infiltration, which recognizes the Fc portion of IgA molecules [93]. Neutrophils expressing c-Kit exhibit enhanced surface expression of proteinase 3 (PR3), the target of anti-neutrophil cytoplasmic antibodies (ANCAs) in vasculitis [94]. The binding of ANCAs to PR3 on these neutrophils can trigger NET formation, contributing to tissue damage in vasculitis [95]. Inhibiting mitogen-activated protein kinase (MAPK) signaling has been proposed as a potential strategy to modulate NET formation (Figure 4) [96]. A recent single-cell RNA sequencing study identified distinct neutrophil subsets in patients with microscopic polyangiitis (MPA). Neutrophils were clustered into seven subsets, with two subsets, immNeus and neutrophils expressing IFN marker genes, found to be elevated in MPA patients. Trajectory and cell-to-cell interaction analyses revealed that IFN-expressing neutrophils formed a distinct primed cluster, which was linked to persistent vasculitis symptoms. Targeting the IFN-γ pathway may thus represent a promising therapeutic approach for MPA (Figure 4) [97]. However, further characterization of these IFN-expressing neutrophils may uncover more selective drug targets.

### 6.4. Systemic Lupus Erythematosus

Systemic lupus erythematosus (SLE) is a chronic autoimmune disease influenced by an interplay of environmental, genetic, and endocrine factors. It is characterized by an overproduction of autoantibodies targeting various self-antigens, resulting in tissue damage [98]. Suppression of ferroptosis with Liproxstatin-1 treatment has been shown to diminish disease progression (Figure 4) [99]. Ferroptosis is an iron-dependent form of regulated cell death induced by over-accumulation of lipid peroxides on cellular membranes, and is highly prevalent in SLE. Li et al. (2021) demonstrated that ferroptosis induced by transcriptional suppression of glutathione peroxidase 4 (GPX4), which uses glutathione to detoxify lipid hydroxyperoxides formed during oxidative stress, represents the main form of neutrophil death in SLE [100]. A key factor in SLE pathogenesis is the role of neutrophils as IFN-producing cells [101]. The predominant subtype of neutrophils found in SLE patients was LDNs, which exhibit an increased ability to form NETs and possess a distinct transcriptional profile compared to NDNs. This profile is characterized by the upregulation of pathways associated with neutrophil activation and responses to Type I IFNs [102]. Discrimination of the LDN subpopulation holds potential for the design of therapies that can modulate specific neutrophil phenotypes, thereby maintaining essential components of neutrophil-mediated innate immunity [103]. Therapeutic strategies aimed at key regulatory nodes of NET formation within LDNs, or those enhancing NET degradation, may offer innovative approaches for SLE management. Several therapies are currently in use or under investigation, including Tofacitinib, a JAK1 and JAK3 inhibitor, which, in a phase Ib/IIa trial, appeared to be well-tolerated in 30 patients with SLE, who exhibited a reduction in circulating LDN levels [104,105]; and Idebenone, an antioxidant, which was able to inhibit NETosis in LDNs, attenuating disease manifestation in a murine SLE model [106]. MDSCs exert protective effects by suppressing B cell activity and promoting IL-10 production [107,108], and Laquinimod treatment has been shown to expand and enhance the suppressive function of MDSCs, suggesting that it has a potential therapeutic role [109]. Additionally, MDSCs are also critical for initiating type I IFN signaling in B cells, significantly contributing to the pathogenesis of SLE [110].

### 6.5. Rheumatoid Arthritis

Rheumatoid arthritis (RA) is a chronic, heterogenous autoimmune disease defined by progressive, symmetric joint inflammation that results in bone erosion, cartilage damage, and functional impairment [111]. Neutrophil numbers are significantly increased in the joints of RA patients, in which elevated circulating levels of MPO, NE, and PAD4 have also been detected [112,113]. These effector pathways can be targeted by selective inhibitors (Figure 4), yet they are not always specific to the pathogenic neutrophil subset(s). In a study, LDNs and MDSCs were detected in the plasma of RA patients. These LDNs exhibit diminished expression of cytokine receptors, such as TNF receptors (TNFRs) [114]. Given that RA pathogenesis is characterized by heightened TNF-α signaling, this observation could potentially elucidate the clinical phenomenon of non-responsiveness to TNF-α inhibitor therapy observed in subsets of RA patients [115,116]. Thus, a potential therapeutic approach could involve directly targeting TNFRs, but no therapies of this type have been developed [117]. MDSCs can inhibit the activation and differentiation of T lymphocytes into Th1 and Th17 cells [118] and promote Tregs expansion, impacting disease progression [119]. Given the demonstrated capacity of exogenous MDSCs to suppress RA in select murine models, these cells present a promising avenue for development as a cellular therapy against RA [119].

### 6.6. Multiple Sclerosis

Multiple sclerosis (MS) is a chronic neurodegenerative disease of the central nervous system (CNS), characterized by immune-mediated demyelination and neuroinflammation. As neutrophil-driven demyelination is linked to the activity of PAD2 and PAD4, selective PAD inhibitors may be a therapeutic strategy for MS patients [120]. Targeting NE activity [121] and pro-inflammatory cytokine production [122] may also represent therapeutic strategies for MS (Figure 4), but these are not directed against specific neutrophil subsets linked to MS pathogenesis. Recent single-cell RNA sequencing of cervical lymph nodes, the primary lymph nodes draining the CNS, in an experimental model of MS, identified three distinct neutrophil subsets with notable migratory potential. The study revealed that this migration was partially driven by Triggering Receptor Expressed on Myeloid cells 1 (TREM1), which promotes AKT activation and NOX2-dependent superoxide production. Additionally, in individuals with MS, there was a significant increase in peripheral neutrophils, accompanied by upregulation of TREM1 expression [123]. Ablation or genetic inactivation of TREM1 has demonstrated significant therapeutic benefits in autoimmune diseases, leading to a marked reduction in inflammatory cell infiltration and a corresponding decrease in pro-inflammatory cytokine expression [124,125]. Pharmacological TREM1 silencing could represent a promising therapeutic approach to MS.

### 6.7. Type 1 Diabetes

Type 1 diabetes (T1D) is a chronic autoimmune disease characterized by insulin deficiency, which results in hyperglycemia. The development of T1D is influenced by a combination of genetic, environmental, and triggering factors, which can be categorized into three distinct stages. Stage 1 is characterized by the presence of autoantibodies in individuals who remain normoglycemic. In stage 2, dysglycemia begins to emerge, and stage 3 represents the onset of symptomatic disease, with hyperglycemia that requires insulin therapy. Neutrophils have been found in the pancreas of T1D patients, and studies have shown a reduction in peripheral blood neutrophil levels in individuals at stages 2 and 3 of the disease [126], probably due to the infiltration of neutrophils into the pancreas in both pre-symptomatic and symptomatic stages. Inhibiting neutrophil recruitment with CXCR2 antagonists has been shown to attenuate diabetic T cell responses, potentially preventing the progression of autoimmune diabetes (Figure 4) [127]. The existence of distinct neutrophil populations in the development of T1D remains unclear. However, RNA sequencing data from neutrophils isolated from subjects at various stages of the disease, as well as from autoantibody-negative subjects with a family history of T1D, have suggested the presence of neutrophil populations with distinct expression patterns of IFN-sensitive genes compared to non-diabetic subjects [128]. To selectively target IFN-expressing neutrophils in the progression of T1D, identifying specific upregulated markers or receptors on these cells could facilitate the development of targeted therapies, such as antibodies or small molecules. These therapeutics could be designed to either modulate or deplete IFN-expressing neutrophils, potentially halting or slowing disease progression. In addition to dysregulated apoptosis, ROS production, and phagocytosis, T1D subjects have been shown to display alterations in NETosis. Some studies have reported elevated levels of NETs, associated with increased circulating NE levels, which may be linked to the initiation of beta cell autoimmunity [129,130]. However, other studies have failed to confirm these dysregulated neutrophil functions across different stages of T1D development [131,132,133]. In a recent study, Bissenova et al. (2023) studied the proteome of NETing neutrophils (NETome) from individuals with T1D, revealing that neutrophils from those with longstanding T1D had fewer proteins associated with innate immunity, irrespective of normal NETosis levels. Moreover, these individuals had neutrophils enriched in metabolic proteins, indicative of adaptation mechanisms utilized by activated neutrophils to avoid compromised glycolysis and ensuing extreme or suboptimal NETosis [133]. These insights might be further exploited to modify specific neutrophil functions in T1D subjects with poor glucose control. The identification of distinct neutrophil populations in the pancreas may further shed light on how to target these first-line defenders in T1D initiation and progression.

### 6.8. Cancer

Cancer is characterized by uncontrolled cell division, driven by both genetic alterations and the external environment, known as the TME (see Section 5.2) [134]. Considering the role of neutrophils in the TME, these cells can be therapeutically targeted to enhance their anti-tumor activity. Several strategies have been explored, ranging from exploiting neutrophil-derived exosomes as carriers of anti-tumor agents, to blocking neutrophil recruitment to the tumor, reprogramming N2 neutrophils into an N1-like phenotype to boost anti-tumor immunity, and modulating neutrophil behavior through the targeting of specific subsets. Neutrophil-derived exosomes show promise as drug carriers, with studies on mice demonstrating the ability of these exosomes to modulate immune responses within the TME, potentially leading to immune evasion by tumors [135]. Their capacity to cross the BBB also makes them a potential tool for treating brain cancer. Another approach involves inhibiting pro-tumor neutrophils with CXCR2 inhibitors and therapies targeting NETs. CXCR2 inhibitors can reduce neutrophil infiltration into the TME and enhance CD8^+^ T cell activation. PAD4, which regulates CXCR2 expression, can be inhibited to limit neutrophil recruitment. PAD4 inhibition, using compounds like Cl-amidine or GSK484, has shown reduced tumor growth and metastasis in animal models. Recently, Zhu et al. (2024) demonstrated that compound 28, a PAD4 inhibitor, hindered PAD4-citH3-NET signaling and suppressed tumor progression [136]. Inefficient antigen cross-presentation in the TME hampers the generation of effective anti-tumor immune responses. Neutrophil reprogramming using nanomaterials or gene editing has been shown to enhance tumor antigen cross-presentation and anti-tumor cytotoxicity. Nanoscale metal–organic frameworks (nMOFs), activated by radiotherapy–radiodynamic therapy, can transform neutrophils into CD11b^+^Ly6G^+^CD11c^+^ hybrid neutrophils with upregulated expression of the costimulatory molecules CD80 and CD86, as well as of MHC class II molecules [137,138,139]. Focusing on neutrophil metabolic pathways may offer another novel strategy to enhance cancer immunotherapy (reviewed in [140]). However, studying neutrophil metabolism remains challenging due to their short lifespan and the TME’s metabolic complexity, requiring advanced research methods. Recently, IFN-stimulated Ly6E^hi^ neutrophils, induced by tumor-intrinsic activation of the STING (stimulator of interferon genes) pathway, were found to sensitize otherwise non-responsive tumors to anti-PD1 therapy. This effect was partly mediated through IL-12b-dependent activation of cytotoxic T cells, highlighting the potential of targeting Ly6E^hi^ neutrophils to enhance immunotherapy efficacy [141]. Additionally, various neutrophil subtypes may exhibit pro-tumor activity. Zhang et al. (2024) found that P2RX1-deficient neutrophils contributed to immunosuppressive effects in non-small-cell lung cancer (NSCLC), suggesting that P2RX1 could be a potential target to counteract the immunosuppressive activity of neutrophils [142]. Wu et al. (2024) generated single-cell neutrophil transcriptomes from seventeen cancer types, revealing ten distinct neutrophil states, including inflammation, angiogenesis, and antigen presentation. The HLA-DR^+^CD74^+^ neutrophil subset was linked to better survival in most cancers, and was triggered by leucine metabolism and H3K27ac modification. The authors found that a leucine diet plus anti-PD-1 therapy significantly reduced tumor volumes [69]. The challenge of selectively targeting pro-tumor neutrophils remains, due to the lack of definitive surface markers, with current identification largely dependent on functional assays. Further investigation into key molecules that regulate or mediate the pro-oncogenic roles of these neutrophils could lead to the discovery of novel therapeutic targets, advancing the development of innovative cancer immunotherapies.

## 7. Conclusions

This review highlights the complexity and functional heterogeneity of neutrophils, challenging the traditional perception of these cells as mere first responders. Single-cell RNA sequencing has revealed distinct subpopulations, yet challenges remain, including technical limitations and a lack of standardized nomenclature. In conclusion, neutrophils are far more than first responders; their development, maturation, and function are shaped by intricate intrinsic and extrinsic factors. Understanding their heterogeneity could enable targeted therapies for infections, cancer, and autoimmune disorders. Future research should refine single-cell technologies and investigate neutrophil–microenvironment interactions to unlock novel therapeutic strategies.

## Figures and Tables

**Figure 1 biomedicines-13-00597-f001:**
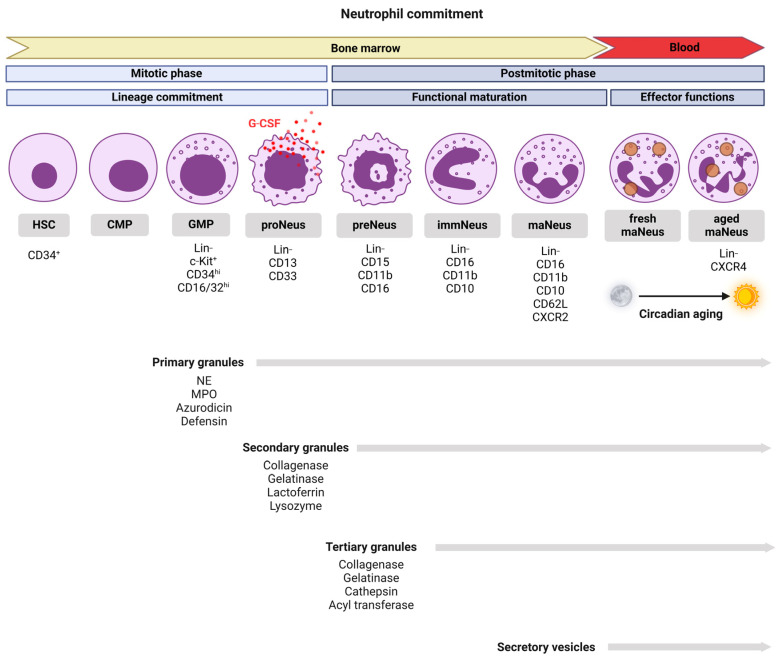
**The stages of neutrophil development, from hematopoietic stem cells in the bone marrow to mature neutrophils in the circulation**. Neutrophil differentiation occurs in two phases: mitotic (hematopoietic stem cells (HSCs) → common myeloid progenitors (CMPs) → granulocyte-monocyte progenitors (GMPs) → pro-neutrophils (proNeus)), and postmitotic (pre-neutrophils (preNeus) → myelocytes → metamyelocytes → immature neutrophils (immNeus) → mature neutrophils (maNeus)). Key lineage markers are indicated. Neutrophils express the C-X-C chemokine receptor CXCR2 when they are freshly released from the bone marrow (fresh maNeus), which helps them to respond to chemokines like IL-8 and migrate to sites of infection or inflammation. As neutrophils age in circulation, they upregulate CXCR4 and downregulate CXCR2 (aged maNeus), promoting their return to the bone marrow for clearance. Neutrophil trafficking follows circadian rhythms, with CXCR2⁺ fresh maNeus released from the bone marrow during the active phase for immune defense, while CXCR4⁺ aged maNeus are cleared back to the bone marrow via the CXCR4-CXCL12 axis during the resting phase, regulated by core clock genes. Neutrophils also contain primary (azurophilic) granules with enzymes like myeloperoxidase (MPO), neutrophil elastase (NE) for microbial killing, secondary (specific) granules with lactoferrin and collagenase for reactive oxygen species (ROS) production, and tertiary (gelatinase) granules with metalloproteinases for tissue migration. Secretory vesicles play a key role in replenishing membrane enzymes, such as alkaline phosphatase and ATPase, by delivering them to the cell surface following cellular activation.

**Figure 2 biomedicines-13-00597-f002:**
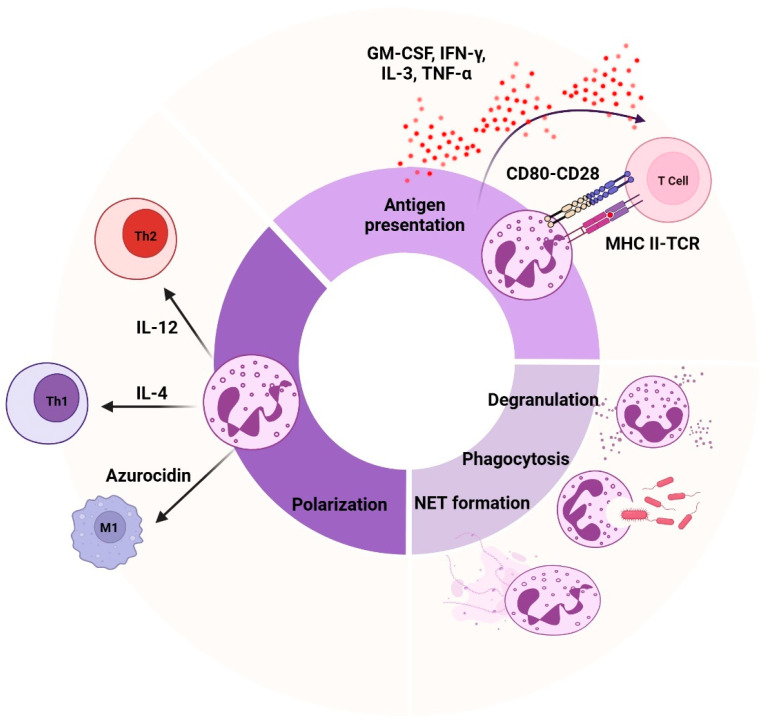
**Neutrophil effector functions.** Neutrophils play a crucial role in immune defense by engulfing pathogens or cellular debris, forming a phagolysosome to degrade the ingested material. They also release anti-microbial granules into the phagosome or the extracellular space to eliminate infections. Neutrophil extracellular traps (NETs) are triggered by inflammatory signals, immune complexes, bacteria, fungi, and protozoa. These web-like structures are composed of decondensed chromatin and granule proteins. NETs can serve two functions: vital, where neutrophils retain their functionality, while releasing NETs to trap and kill pathogens; or suicidal, where neutrophils die and release both their intracellular contents and the NETs. Neutrophils can behave as antigen-presenting cells, directly interacting with T cells through MHC II-TRC binding, CD80-CD28 interaction, and the secretion of GM-CSF, IFN-γ, IL-3, and TNF-ɑ. Additionally, neutrophils can influence T cell and macrophage polarization by secreting cytokines and granule content. IL-12 promotes Th1 differentiation, IL-4 favors Th2 differentiation, and azurocidin promotes M1 macrophage proliferation.

**Figure 3 biomedicines-13-00597-f003:**
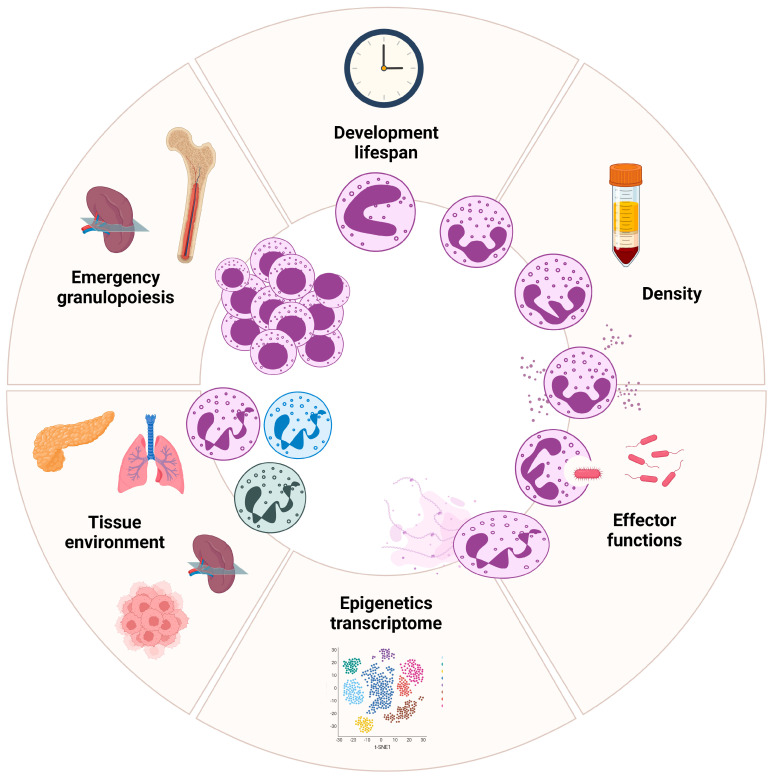
**Neutrophil plasticity and transcriptional heterogeneity during maturation and tissue adaptation.** Neutrophils go through multiple differentiation stages in the bone marrow, with each stage displaying unique transcriptional profiles. The lifespan of neutrophils is influenced by circadian clocks, with their activity and survival varying throughout the day, typically peaking during the night. Neutrophils exhibit distinct phenotypic and functional states, shaped by both intrinsic and extrinsic factors, in steady-state and disease contexts. Their heterogeneity extends to peripheral blood and tissues, where they specialize in response to environmental cues. Epigenetic and transcriptomic analyses have revealed that this variability reflects functional plasticity, suggesting that neutrophil subsets may not be entirely distinct, but may dynamically adjust based on stimuli.

**Figure 4 biomedicines-13-00597-f004:**
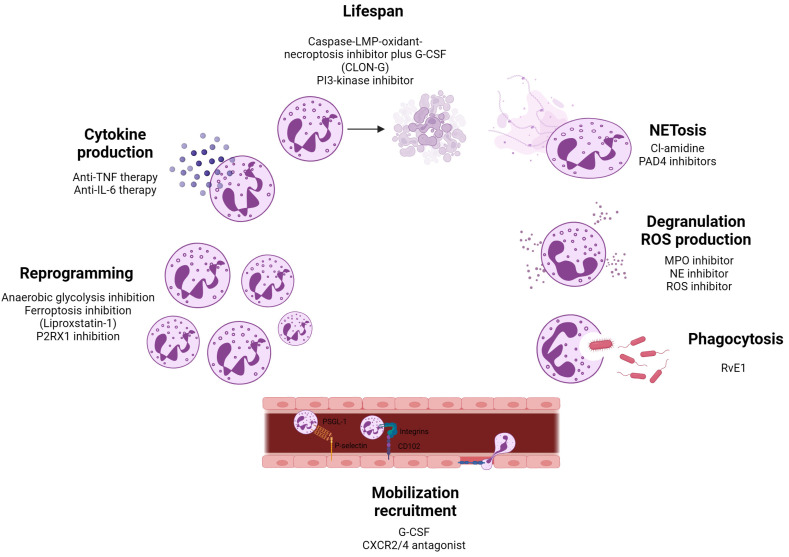
**Therapies targeting neutrophils and their functions.** This figure summarizes therapeutic strategies aimed at modulating key neutrophil effector functions, including lifespan regulation, NETosis, degranulation, ROS production, phagocytosis, mobilization, recruitment, reprogramming, and cytokine production. The latter can be attenuated by anti-TNF and anti-IL-6 therapies. Targeting neutrophil lifespan can involve treatment with caspases-lysosomal membrane permeabilization-oxidant-necroptosis inhibition plus granulocyte colony-stimulating factor (CLON-G) or PI3-kinase inhibitors. NETosis can be inhibited using Cl-amidine and protein-arginine deiminase (PAD)4 inhibitors. Degranulation and excessive ROS production can be suppressed using selective inhibitors for myeloperoxidase (MPO), neutrophil elastase (NE), and ROS production. Phagocytic capacity can be enhanced via resolvin E1 (RvE1). Neutrophil mobilization and recruitment can be regulated by G-CSF and CXCR2/4 antagonists, providing potential avenues for therapeutic intervention in inflammatory and immune-mediated diseases. Reprogramming strategies, including inhibition of anaerobic glycolysis, of ferroptosis, or of the purinergic receptor P2RX1, aim to modulate neutrophil metabolism, non-apoptotic cell death, and activation.

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
