# Peer review of "Exploring Neutrophil Heterogeneity and Plasticity in Health and Disease"

_biomedicines, 2025, doi:10.3390/biomedicines13030597_

Round 1

Reviewer 1 Report

Comments and Suggestions for Authors

The review by Gysemans and co-authors deals with the role of neutrophils in physiological and pathological conditions, with a particular focus on cell heterogeneity that encompasses different phenotypic and functional traits. As a result of this effort, the manuscript covers a wide range of topics, with both basic information for introduction to each topic as well as up-to-date developments. In general, the manuscript is well written and the figures clearly summarize the major topics of the manuscript. I have few comments that authors may want to address:

1- The initial part of section 2 covers the development of neutrophils, a topic that is discussed also in section 5.1. It is not clear to me why the topic is not covered in a single section.

2- in my opinion, some descriptions appear out of focus and could be omitted or reduced, for instance the description of neutrophil recruitment (lines 91-112), the description of the granules (lines 192-202), or the description of phagocytosis, degranulation and NETosis (lines 238-257)

3- although mostly specified in the text, it is sometimes unclear whether the description refers to human or mice, for instance the role of neutrophils in antigen presentation

4- The link between the heterogeneity of neutrophils discussed in sections 1-5, and the diseases and therapeutic approaches described in section 6, is not always clear to me, for instance in the description of sepsis or gout. I was expecting a focus on targeting neutrophil heterogeneity, rather than general effector functions, as therapeutic strategy, which may be more in line with the scope of the manuscript.

Minor:

- lines 98-99: the sentence is unclear to me (what binds endothelial selectins?)

- Figure 1: secretory vesicles should be included

- refs 25 and 26 are switched

- line 524: please, correct 20323

Author Response

Reviewer: 1

I have a few comments that authors may want to address:

  1. The initial part of Section 2 covers the development of neutrophils, a topic that is also discussed in Section 5.1. It is not clear to me why the topic is not covered in a single section.

In Section 2, we primarily discussed the general view of neutrophil development, while Section 5.1. goes into details into the epigenetic and transcriptional regulation of neutrophil development from bone marrow to the circulation. The exploitation of novel technologies like multi-omics analyses just lately provided more in-depth insights into how neutrophil subsets are shaped by intrinsic and extrinsic factors. We shortened the preface of this section to immediately move to the recent advances offered by single-cell omics studies.

  1. In my opinion, some descriptions appear out of focus and could be omitted or reduced, for instance the description of neutrophil recruitment (lines 91-112), the description of the granules (lines 192-202), or the description of phagocytosis, degranulation, and NETosis (lines 238-257).

We intended to provide a broad description of the different neutrophil functions so not only experts in the field but also readers with less neutrophil background receive a complete view of these features. We made some parts more comprehensive and referred to outstanding reviews on the topics.

  1. Although mostly specified in the text, it is sometimes unclear whether the description refers to human or mice; for instance, the role of neutrophils in antigen presentation

We have gone over the text and adapted parts to better make a distinction between mouse and human data (indicated in yellow). However, in the part on antigen presentation, we clearly mentioned that both in the mouse and human context, neutrophils with antigen presentation features have been detected (page 8).

  1. The link between the heterogeneity of neutrophils discussed in sections 1–5 and the diseases and therapeutic approaches described in section 6 is not always clear to me, for instance in the description of sepsis or gout. I was expecting a focus on targeting neutrophil heterogeneity rather than general effector functions as a therapeutic strategy, which may be more in line with the scope of the manuscript.

The reviewer is correct that the last part mainly focused on how therapeutic strategies could impact specific neutrophil (effector) functions and less neutrophil heterogeneity. As this field is very novel, only a few papers addressed this approach. Yet, we modified the last part to align better with the general focus of the review. We substantially shorted this part yet prioritized therapeutic strategies targeting neutrophil subsets in inflammatory, autoimmune, and cancer contexts.

Minor:

  1. lines 98-99: the sentence is unclear to me (what binds endothelial selectins?)

We have adapted this part on neutrophil recruitment to make it more understandable.

  1. Figure 1: secretory vesicles should be included

Secretory vesicles are now included in figure 1 (and text).

  1. refs 25 and 26 are switched

References were wrongly integrated and corrected.

  1. line 524: Please correct 20323

We have corrected this typo.

Reviewer 2 Report

Comments and Suggestions for Authors

This review underscores the complexity and functional heterogeneity of neutrophils, challenging the traditional view of these cells as mere first responders. The distinction between HDNs and LDNs further highlights this heterogeneity. Overall, this literature review summarizes an expansive group of articles surrounding neutrophil plasticity and presents the information in a systematic, logical manner. The information is easily digestible and provides source material for easy access to further studies.

Major:

1, Line 766, Section 6.9: This section does not extensively discuss neutrophil heterogeneity and plasticity.

2, the conclusion is quite comprehensive and well-written, but it is on the longer side for a review paper. Generally, conclusions in reviews should be concise and focused, summarizing key findings without excessive detail.

Minor:

1, Line 32, it's better to clarify the numerical range to avoid ambiguity. The current format, "5 × 10¹⁰-10¹¹", might be misinterpreted.

2, Line 80, singular/plural agreement “its”

3, Line 85, "band neutrophils (bandNeus)" for clarity

4, Line 85 define “maNeus”

5, Line 211: Is there any other research regarding HDN and LDN populations?

Author Response

Major:

  1. Line 766, Section 6.9: This section does not extensively discuss neutrophil heterogeneity and plasticity.

As mentioned to reviewer 1, we have revised this section to have the focus on neutrophil heterogeneity and less on neutrophil functions.

  1. The conclusion is quite comprehensive and well-written, but it is on the longer side for a review paper. Generally, conclusions in reviews should be concise and focused, summarizing key findings without excessive detail.

We have substantially shortened the conclusion, yet maintaining the focus on novelties in the field of neutrophil heterogeneity and plasticity and how these can be exploited from a therapeutic point of view.

Minor:

1, Line 32, it's better to clarify the numerical range to avoid ambiguity. The current format, "5 × 10¹⁰-10¹¹", might be misinterpreted.

We adapted this numerical range in the text to avoid confusion.

2, Line 80, singular/plural agreement "its.”.

We have corrected this mistake.

3, Line 85, "band neutrophils (bandNeus)" for clarity.

We have integrated the band neutrophils.

  1. Line 85 defines “maNeus” already.

maNeus were defined earlier in the text.

  1. Line 211: Is there any other research regarding HDN and LDN populations?

We have provided more insights into these HDNs and LDNs by integrating some reviews covering this topic.

Reviewer 3 Report

Comments and Suggestions for Authors

Well written manuscript providing a very clear insight on the role of neutrophil heterogeneity in healthy, disease and therapeutics

Authors should ensure that abbreviations are stated in long form the first time they are used

The manuscript can be accepted once authors have addressed the small concern of abbreviations

Author Response

  1. Authors should ensure that abbreviations are stated in long form the first time they are used. The manuscript can be accepted once authors have addressed the small concern of abbreviations.

We have gone over the manuscript and integrated the long form of the abbreviations used in the text.
